# Targeted Sequencing of Cytokine-Induced PI3K-Related Genes in Ulcerative Colitis, Colorectal Cancer and Colitis-Associated Cancer

**DOI:** 10.3390/ijms231911472

**Published:** 2022-09-29

**Authors:** Nurul Nadirah Razali, Raja Affendi Raja Ali, Khairul Najmi Muhammad Nawawi, Azyani Yahaya, Norfilza M. Mokhtar

**Affiliations:** 1Department of Physiology, Faculty of Medicine, Universiti Kebangsaan Malaysia, Cheras, Kuala Lumpur 56000, Malaysia; 2Gastroenterology Unit, Department of Medicine, Faculty of Medicine, Universiti Kebangsaan Malaysia, Cheras, Kuala Lumpur 56000, Malaysia; 3GUT Research Group, Faculty of Medicine, Universiti Kebangsaan Malaysia, Cheras, Kuala Lumpur 56000, Malaysia; 4Department of Pathology, Faculty of Medicine, Universiti Kebangsaan Malaysia, Cheras, Kuala Lumpur 56000, Malaysia

**Keywords:** targeted sequencing, inflammatory bowel diseases, colorectal cancer, colitis-associated cancer, phosphatidylinositol 3-kinase

## Abstract

Chronic relapsing inflammatory bowel disease is strongly linked to an increased risk of colitis-associated cancer (CAC). One of the well-known inflammatory carcinogenesis pathways, phosphatidylinositol 3-kinase (PI3K), was identified to be a crucial mechanism in long-standing ulcerative colitis (UC). The goal of this study was to identify somatic variants in the cytokine-induced PI3K-related genes in UC, colorectal cancer (CRC) and CAC. Thirty biopsies (n = 8 long-standing UC, n = 11 CRC, n = 8 paired normal colorectal mucosa and n = 3 CAC) were subjected to targeted sequencing on 13 PI3K-related genes using Illumina sequencing and the SureSelectXT Target Enrichment System. The Genome Analysis Toolkit was used to analyze variants, while ANNOVAR was employed to detect annotations. There were 5116 intronic, 355 exonic, 172 untranslated region (UTR) and 59 noncoding intronic variations detected across all samples. Apart from a very small number of frameshifts, the distribution of missense and synonymous variants was almost equal. We discovered changed levels of *IL23R*, *IL12Rß1*, *IL12Rß2*, *TYK2*, *JAK2* and *OSMR* in more than 50% of the samples. The *IL23R* variant in the UTR region, rs10889677, was identified to be a possible variant that might potentially connect CAC with UC and CRC. Additional secondary structure prediction using RNAfold revealed that mutant structures were more unstable than wildtype structures. Further functional research on the potential variants is, therefore, highly recommended since it may provide insight on the relationship between inflammation and cancer risk in the cytokine-induced PI3K pathway.

## 1. Introduction

Carcinogenesis is the most severe complication that could arise from prolonged inflammatory bowel disease (IBD). Colitis-associated cancer (CAC) is a type of colorectal cancer (CRC) that develops as a consequence of IBD due to the presence of chronic inflammation in the gastrointestinal tract [1,2]. The likelihood of CAC developing from ulcerative colitis (UC) and Crohn’s disease (CD), the two major subtypes of IBD, is 1.4% and 0.8%, respectively [3,4]. Additionally, the prevalence rate of developing CAC is expected to range from 0.6 to 17% in Western nations and from 0.3 to 1.8% in Asia Pacific regions [5]. Moreover, in Malaysia, the mean incidence of IBD doubled to 1.46 per 100,000 person/year in between 2010 and 2018 [6]. The rising urbanization of societies, which includes dietary changes, the use of antibiotics, personal cleanliness standards, microbiological exposures, and pollution, may be the cause of the rising trend in IBD over the past decades [7]. As a result, this increasing trend may ultimately contribute to the dynamic shift of CAC among Asians. Moreover, CAC has contributed to 10 to 15% of IBD fatality cases in Western countries [4,8].

Generally, CAC only accounts for 1 to 2% of CRC cases [8]. There are several characteristics that may distinguish CAC from sporadic CRC. Their clinicopathological characteristics are comparable, but CAC has a higher proportion of numerous cancer lesions, an increased percentage of superficial and invasive type lesions, and a higher proportion of mucinous or signet ring cell carcinomas [9]. Contrary to CRC, CAC develops through the inflammation–dysplasia–carcinoma sequence, where the change from low to high grade dysplasia is triggered by field precursor cells that are present in or close to the dysplastic mucosa [10]. According to a study by Choi et al. (2015), 20% of UC patients with low-grade dysplasia may have developed high-grade dysplasia or CRC within 53 months of their first diagnosis [11].

Interestingly, there are many similarities between the molecular pathogenesis of CAC and CRC. *K-Ras*, *p53*, *APC* and *COX2* are some of the common genes that are altered as CAC and CRC progressed [8]. As p53 is widely distributed in the inflamed mucosal area, this suggests that chronic inflammation has a propensity to become mutagenic [12]. In fact, a study by Claessen et al. (2010) indicated that *p53* staining was found moderately in non-dysplastic tissue and exhibition of stronger expression was seen in the low- and high-grade dysplastic lesion in more than 60% of IBD patients [13].

Immune cells, epithelial cells, stromal cells, cytokines, and chemokines are among the diverse cell types that make up the inflammatory process in CAC development and are similar to those found in the microenvironment of the malignancy [14]. Cytokines have a tendency to control the pro-tumorigenic response in chronic inflammatory conditions by causing cell malignancies and transformation [15,16]. Inflammatory mediators such as tumor necrosis factor alpha (*TNF-α*), interleukin-6 (*IL6*) and *STAT3* played significant roles in pre-neoplastic growth regulation during CAC tumorigenesis as demonstrated in an animal model study [17]. 

The release of those mediators is more likely to target several signaling pathways that play major roles in carcinogenesis such as NF-κB, PI3K, JAK/STAT and Wnt/B-catenin [18]. Phosphatidylinositol 3-kinases (PI3K) were recognized in promoting cancer progression as they play a key role in the regulation of survival, differentiation, and proliferation of cancer cells. PI3K enzymatic activity was found to be involved in the pathogenesis of various diseases, ranging from chronic inflammation to cancer, for instance CRC [19,20]. A recent microarray study on UC patients with two different durations discovered PI3K as one of the important pathways in the long-duration UC compared to short duration [21]. Nevertheless, there is still a paucity of knowledge about the role of the PI3K signaling pathway in the carcinogenesis progression of colitis-associated cancer. Thus, in the present study, we have performed targeted sequencing on 13 genes that are related to the cytokines-induced PI3K signaling pathway for the identification of driver gene mutations in colitis-associated cancer, long-standing ulcerative colitis, and sporadic colorectal cancer patients.

## 2. Results

### 2.1. Information on Clinical Samples

Table 1 displays demographic data for all samples. The median age of all samples was 69 years old (IQR:8.75). Malays made up the majority of the samples (70%) and were followed by Chinese (17%) and Indians (13%). Females had a somewhat larger gender distribution (60%) than men (40%). Most patients’ smoking status was non-smoker (93%) as opposed to ex-smoker (7%). The mean disease duration for all long-standing UC was 28.5 ± 6.61 years. Most UC patients had a diagnosis of left-sided colitis or pancolitis, with a Mayo index score of 1 to 3 and a Geboes score of Grade 2A.1 to 2A.2 of. Meanwhile, patients with CAC had a chronic active history of colitis for 22 ± 13 years. The majority of CRC and CAC patients were at stages 1 to 3, and the rectosigmoid and distal colon were the sites of the malignancies. Histologically, the majority of CRC were moderately differentiated, whereas tumors from CAC patients were categorized as poorly and well differentiated. None of the patients had a history of CRC in their families.

### 2.2. The Technical Performance of the Target Enrichment System Panel

The average percentage of clean reads across all raw reads produced by the SureSelectXT Target Enrichment System was 96.4% (range 80.9 to 98.8%). The range of the total reads was 1,200,800 to 6,106,992 reads. For each sample, the percentage of the target base covered by at least of 100× for each sample had reached 100%. 

### 2.3. Summary of Identified Variants in PI3K-Related Genes 

Targeted sequencing was performed on 13 genes that are related to PI3K, namely *IL12Rß1, IL12Rß2, IL23R, IL31, OSMR, JAK2, TYK2, STAT1, STAT3, STAT4, STAT6, PDK1* and *SGK2*. Long-standing ulcerative colitis (n = 8), colitis-associated cancer (n = 3), colorectal cancer (n = 11) and paired normal colorectal mucosa (n = 8) made up the total 30 samples. In total, we found 5702 variants across all samples in the cytokine-induced PI3K-related genes. Ninety percent (5116) of such variants were intronic mutations, followed by 355 exonic mutations (6%), 172 mutations in the 3′ and 5′ untranslated region (UTR) region (3%) and 59 mutations on the noncoding intronic region (1%) (Figure 1A).

Single-nucleotide polymorphisms (SNPs) made up most of the discovered variants (75% = 4256 variants), followed by 25% (1446 variants) of insertion–deletion (InDel) variants. The top three genes with the most SNPs were *OSMR* (640 variants), *IL12Rß1* (549 variants) and *IL23R* (515 variants). Meanwhile, *IL12Rß2* (202 variants), *IL12Rß1* (174 variants) and *STAT6* (163 variants) showed a markedly high frequency of number of InDel variants (Figure 1B).

The number of missense and synonymous mutations from a total of 355 exonic variants had somewhat similar distribution, with 179 variants (50.4%) and 171 variants (48.2%), respectively, followed by 5 variants (1.4%) with frameshift mutations (insertion and deletion). Only 5.6% (n = 10) of missense mutations in the four genes *IL12Rß1, OSMR, JAK2* and *STAT1* were predicted to be damaging or potentially damaging, whereas the remaining 174 variants (94.4%) were predicted to have a benign or neutral function (Figure 1C). 

More than 50% of the patients had changed exonic sequences in 6 out of the 13 cytokine-induced PI3K-related genes. *IL23R* missense mutations appeared in all samples including the normal colonic mucosa, which is the paired sample of CRC patients (30/30 samples). This was followed by *IL12Rß2*, where all paired normal samples had the same mutations as CRC (28/30 samples). In contrast, missense mutations *TYK2* and *OSMR* were found in 27 and 21 out of 30 samples, respectively, even though 6 of those samples had paired normal colonic tissues. A total of 7 out of the 30 samples were matched normal samples where 23 of them contained missense mutations in the *JAK2*. Meanwhile, *IL12Rß1* missense mutations were present in 19 out of 30 samples, including four matched normal tissues. Apart from *IL12Rß2* and *JAK*2, where synonymous mutations were frequently observed, practically all top changed genes showed significant amount of missense mutation. Frameshift mutations, however, were infrequently observed and were only found in *OSMR* (2/21 samples) and *TYK2* (1/27 samples) (Figure 2).

### 2.4. Somatic Variants Distribution in PI3K-Related Genes among All Samples

A total of 634 mutations were found as recurrent mutations, in the most frequently mutated gene, *IL23R*. Almost half of the 314 overall mutations were discovered in two or more samples. Of the 634 total mutations, 57 were recurrent missense mutations, 3 were synonymous mutations, 551 were intronic and 23 were UTR. These mutations were discovered sporadically in exon 2 to 10 (Figure 3A).

Only 453 alterations with 102 recurrent mutations were found in the second-most frequently occurring gene, *TYK2*, which had a 90% incidence of mutations. A total of 362 intronic and 3′-5′ flanking regions, 31 missense, and 6 synonymous were discovered throughout exon 1 to 23. *TYK2* had the most splice mutations, 50 in total (Figure 3B).

Meanwhile, *IL12Rß2* reported 489 alterations with recurring 163 mutations. Among all groups studied, 60 recurrent synonymous mutations were distributed across exons 2 to 15, followed by 416 intronic and 12 splice areas. In addition, exon 10 only contained one missense mutation (Figure 3C). 

*IL12Rß1* had the highest number of alterations, with 723 mutations, including 299 recurrences while being affected in just 60% of all cases. Between exon 7 to 15, 55 recurrence missense mutations were discovered, while 28 synonymous mutations were widely distributed around exon 1 to 16, followed by 623 intronic and 8 UTR (Figure 3D). 

Another gene with a lot of modification numbers is *OSMR*, with 699 alterations, 281 of which were recurrent mutations. Exon 2 to 18 had 641 intronic mutations, 28 UTR, 20 missense mutations, and 8 synonymous mutations. Two samples contained a single frameshift insertion in exon 10, resulting in the protein’s function being truncated (Figure 3E).

On the other hand, *JAK2* has 120 recurring mutations with 461 overall alterations. Throughout exonic region 3 to 25, 34 synonymous, 20 3′UTR, 400 intronic and 2 splice mutations were found in exonic region 3 to 25. Additionally, at exon 9 and 17, only two missenses were discovered. Exon 8 and 13 both had two frameshift deletion mutations that led to truncated proteins (Figure 3F).

However, no mutations were identified in the cancer hotspot locations of *IL23R*, *IL12Rß1*, *IL12Rß2*, *OSMR*, *TYK2*, and *JAK2* across all mutant distributions.

### 2.5. Somatic Variants Distribution of PI3K-Related Genes per Group

The three main groups in this study are ulcerative colitis (UC), colorectal cancer (CRC) and colitis-associated cancer (CAC). As a result, each group’s distribution of variants in PI3K-related genes was also examined.

A total 1625 variants, including 51 missense, 46 synonymous, 1 frameshift, 1461 intronic, 48 UTR and 18 non-coding intronic, were observed in PI3K-related genes in the UC group. The top five altered genes were *IL12Rß2*, *IL23R*, *OSMR*, *STAT1* and *STAT3* (Figure 4A). Additionally, at least two UC samples were the only ones to contain all 26 recurrent mutations in *IL12Rß1*, *IL12Rß2*, *IL23R, SGK2*, *OSMR*, *STAT4* and *STAT6*. All recurrences, however, were only discovered in the intronic region (Table 2).

In the CRC group, 2029 variants were found, including 68 missense, 61 synonymous, 3 frameshift, 1809 intronic, 68 UTR and 20 non-coding intronic variants. The top five altered genes in the CRC group comprised *IL12Rß1*, *IL23R*, *OSMR*, *STAT1* and *STAT3* (Figure 4B). Furthermore, at least two samples from the CRC group only had 19 recurrent mutations in the exonic, intronic, and UTR regions of *IL12Rß1*, *IL12Rß2*, *OSMR*, *JAK2*, *IL23R*, *STAT4* and *STAT6*. Both missense mutations that affected the exonic region were predicted to be benign or neutral (Table 2).

In all, 614 variants were found in the CAC group, comprising 16 missense, 20 synonymous, 561 intronic, 13 UTR and 4 non-coding intronic mutations. Most alterations were found in *IL12Rß1*, *IL12Rß2*, *IL23R*, *OSMR* and *TYK2* (Figure 4C). Only two recurrence frameshift mutations in *IL12Rß2* and *TYK2* were discovered in at least two samples of CAC, in contrast to the UC and CRC groups (Table 2).

We were able to identify variants that co-occurred in the CAC group with the UC and CRC, respectively. Data filtration was applied on the selection of only CAC group, paired with either the UC or CRC group which have yielded a total of 27 intronic variants. For CAC with UC, these co-existing variants were scattered from *IL12Rß1*, *IL12Rß2*, *IL23R*, *OSMR*, *JAK2*, *TYK2*, *STAT1*, *STAT3* and *STAT6*, but for CAC with CRC, only two genes were involved (*JAK2* and *STAT4*) (Table 3).

### 2.6. Identification of Potential Variants That Link UC, CRC and CAC

Additional research was also carried out to identify possible polymorphisms that link CAC with UC and CRC. In particular, we were looking for variants that should be present in the majority of CAC samples. Data were filtered with a high CAC number (minimum n = 2), and low paired normal number (maximum is three out of eight) for better analysis. Data on UC and CRC numbers, however, were not filtered. In addition, because the analysis’s objective was to detect in the coding region, data on the variants’ function were also sorted by ‘exonic’ and ‘UTR’ exclusively.

With the help of data filtering, we identified six potential variants that were present in at least two CAC samples, three in matched normal samples and half of the CRC and UC samples. In *IL12R**ß1*, *IL12R**ß2* and *IL23R*, one missense, four synonymous and one UTR variant were found. Sanger sequencing was used to confirm these possible variants further. Only the variant rs10889677 (c.*309C>A), located at the UTR region of *IL23R,* was validated, and found in practically all CAC samples.

Other than missense mutations, it was not possible to apply protein function prediction methods such as SIFT and PolyPhen-2. Nevertheless, based on their mRNA secondary structure, predictions about the effects of those synonymous and UTR variations are still possible. In terms of the enhanced presence of hairpins, stem-loops, bulge loops, multi-branch loops, and stacking, changes in the secondary structure of the mRNA could be seen.

In silico prediction analysis showed obvious changes in the secondary structure *IL23R* variant (rs10889677) (Figure 5). In comparison to the wildtype, the presence of variant rs10889677 significantly altered the structure close to the terminal branch by adding additional branches and loops. The minimum free energy (MFE) value predicted from the created structure was also affected because of the changing of the color coding on the base structure.

## 3. Discussion

The risk of contracting colitis-related cancer rose in UC patients who had chronic inflammation that persisted for an extended period of time. The development of CAC was recently studied in relation to *p53*, *APC* and *K-Ras* [25]. However, the underlying role of the inflammation–carcinogenesis pathway in the pathogenesis of CAC is still poorly understood. We are interested in investigating how the PI3K signaling pathway contributes to the pathophysiology of CAC. In this study, we used the targeted sequencing approach via the SureSelectXT Target Enrichment System to test for somatic mutations in 13 cytokine-induced PI3K-related genes in long-standing UC, CAC and CRC patients, instead of choosing the common cancer-related genes.

Screening on the somatic alterations in our group samples showed that the majority of the variants were found on the intron, not the exonic region. In fact, silent mutations and neutral predicted variants predominated in the coding region. Few insertions and deletions that could result in frameshift mutations were seen. Nevertheless, the SureSelectXT Target Enrichment System is professed as an accurate genome analysis method from small-scale research to large sample cohorts. It has demonstrated high performance, as measured by capture efficiency, sensitivity, reproducibility, and SNP detection [26]. In this study, this application has successfully sequenced more than 95% of the clean reads coverage with an average error rate of less than 0.15% across all bases. In fact, the high precision of genome sequencing was explained by the fact that there are at least 100× as many reads per given nucleotides in the genome. 

In each of our sample populations, interleukin-23 receptor (*IL23R*) was found to be the gene that was most frequently altered. A previous study has reported *IL23R* as a gene associated with inflammatory bowel disease (IBD), whereby nine SNPs in *IL23R* at various locations such as intronic, exonic and UTR have shown significant associations with Crohn’s disease [27]. Moreover, it has been shown that *IL23R* variants in IBD may operate as a protective variant or contribute to the development of inflammation [28,29]. It has been demonstrated that *IL23R* polymorphisms also may increase the risk of CRC [30]. 

There are several *IL23R* variants that we have discovered, but just two of them, rs7530511 (c.929T>C: p.L310P) and rs1884444 (c.9G>T: p.Q3H), have been linked with colorectal-related diseases. In contrast, both variants were found in intracerebral hemorrhage and cancer-related disease (bladder and esophageal) [31,32,33]. Despite this, one interesting variant in *IL23R*, rs10889677 (c.*309C>A), was primarily discovered in UC, CRC and CAC. This is in line with recent studies that found this polymorphism to be a strong predictor of CRC in Asians and a risk factor for IBD [34,35]. In fact, according to a prior genome-wide association study, rs10889677 demonstrated a favorable link with IBD and might subsequently make the condition worse clinically [27]. Additionally, *IL23R* has just recently been discussed as a potential IBD treatment [36,37]. The activity of the IL23 signaling pathway could be inhibited by specific binding of the oral peptide (PTG-200) to the *IL23R*, which would subsequently influence *JAK2* and *TYK2* activation and perhaps result in abnormal *STAT3* and *STAT4* expression. 

The next frequently altered gene in our study was interleukin-12 receptor beta 1 (*IL12Rß1*). Twenty percent of our samples included the *IL12Rß1* variant (rs11575935; c.1573G>A: p.A525T) that may be harmful. This variant has not yet been connected to any illnesses. Only 1.7% and 0.4% of the population in Asian and European regions, respectively, had this variant [38]. The next frequently altered gene in our study was tyrosine kinase-2 (*TYK2*). *TYK2* has lately gained attention as a potential therapeutic for IBD; its connection to gastrointestinal disorders has long been established [39,40]. *TYK2* mutation rs2304256 (c.1084G>T: p.V362F), which was discovered in almost 80% of our samples, was an intriguing finding. This variant has been linked to autoimmune and inflammatory diseases, including IBD [41,42]. The mutation rs2304256 was initially thought to be benign, but subsequent studies showed that it might encourage exon 8 inclusion and subtly boost *TYK2* expression in whole blood [43]. In fact, the Genotype-Tissue Expression (GTEx) database shows that rs2304256 is indeed linked to a slight increase in *TYK2* in several tissues, including colonic tissue. The Oncostatin M receptor (*OSMR*) is a different gene that has drawn attention. Our finding on the *OSMR* variant, rs2278329 (c.1657G>A: p.D553N), which predominately affects CRC and UC, is analogous to prior studies on *OSMR* that looked at the role of *OSMR* in inflammation and its potential link with other cancers such as bladder and thyroid cancer [44,45,46].

Apart from that, there was only one variant in the untranslated region (UTR) region of *IL23R*, rs10889677, which was found in 85% of CAC samples, and validated as the potential variant that correlates CAC with UC and CRC. Although, the selected potential variant was not located in the coding region and predicted as damaging, yet the effect of that variant on the translation efficacy still can be predicted via the construction of the mRNA secondary structure. The terminal branch of the *IL23R* variant rs10889677’s mRNA secondary structure was clearly altered. Compared to the wildtype structure, prominent additional branches and loops had impacted the minimum free energy (MFE) value. MFE demonstrates the stability of a structure. In a stable structure, there should be more negative values. Stability is essential in the mRNA secondary structure because stable complexes may increase translation efficiency [47]. This finding was also supported by another studies that showed the presence of a non-damaging intronic variant had caused alteration in the mRNA secondary structure and stability, thus affecting the mRNA expression level in the brain tissue [48]. Moreover, results of additional studies also corroborated this assumption, where the mutant rs10889677 variant that had aberrant translation efficiency showed lower rates of T-cell proliferation, subsequently increasing susceptibility of the IBD and elevating the risk for developing cancers, such as breast, lung and nasopharyngeal [49,50]. 

There are a few limitations of this study. This study only involved a single center. Most likely, it was due to our stringent inclusion and exclusion criteria. Moreover, we were cautious in selecting the potential CAC variants, as data filtering was conducted based only exonic and UTR variants. Therefore, further studies with a larger cohort would be highly recommended to corroborate the results. In addition, exploring the underlying mechanism of more variety of potential variants in CAC via in vitro functional study would be highly beneficial in discovering the linkage of those gene variants with tumorigenesis.

Through our findings, we successfully identified somatic variants in cytokine-induced PI3K-related genes in long-standing UC, CAC and CRC samples. Targeted therapies for CRC now focus on a few common pathways, including EGFR (cetuximab and panitumumab) and VEGF (bevacizumab), which can stimulate a number of downstream intracellular signaling pathways including the PI3K signaling pathway [51]. So far, therapeutics interventions based on cytokine-induced PI3K-related genes such as *IL23R* have primarily been studied in inflammatory illnesses [37,52]. Hence, introducing *IL23R* as the next cancer small-molecule inhibitor may be advantageous for future therapies.

## 4. Materials and Methods

### 4.1. Sample Collection

A total of 30 fresh frozen and archive samples from long-standing UC, CAC, CRC and the corresponding adjacent normal colorectal mucosa tissue were collected from patients that were attending the Endoscopy Unit, Universiti Kebangsaan Malaysia Medical Centre (UKMMC), Kuala Lumpur, Malaysia. Upon admission, informed consent was obtained from each patient. Basic clinical and demographic data were gathered and analyzed while reviewing the patient’s medical records. Prior to further processing, the tissues were collected in RNAlater (Sigma Aldrich, St. Louis, MO, USA) and kept frozen at −80 °C. An experienced pathologist examined the confirmation of the diagnosis, the level of inflammation, and the presence of metastases based on the hematoxylin and eosin (H&E)-stained sections. Only cancer tissues with more than 80% tumor cell content were used in this study for cancer samples. The normal samples were confirmed to be free from tumor or inflammatory cells. The Universiti Kebangsaan Malaysia Research Ethics Committee (UKM/PPI/111/8/JEP-2019-572) granted approval for this study.

### 4.2. Nucleic Acid Extraction and Quality Assessment

DNA extraction from the fresh frozen tissues was performed using the AllPrep DNA/RNA/miRNA Universal Kit (Qiagen, Valencia, CA, USA) in accordance with the manufacturer’s protocol. Meanwhile, using GENEREAD DNA FFPE Kit (Qiagen, Valencia, CA, USA), DNA was extracted from formalin-fixed paraffin-embedded (FFPE) blocks of archival samples. DNA concentration was measured using DeNovix DS11+ Spectrophotometer (DeNovix Inc., Wilmington, DE, USA) and Qubit^®^ DNA Assay Kit in Qubit^®^ 2.0 Fluorometer (Life Technologies, Carlsbad, CA, USA). The agarose gel electrophoresis was used to evaluate the extracted DNA’s purity. Targeted sequencing was performed on the genomic DNA that was largely undamaged and free of RNA. 

### 4.3. Targeted Sequencing

Library preparation was conducted using DNA random fragmentation by sonication (Covaris, MA, USA) to the size of 180–280 bp fragments, followed by PCR enrichment and purification with AMPure XP system (Beckman Coulter, Beverly, USA). The library was then quantified using the high-sensitivity DNA assay on the Agilent Bioanalyzer 2100 (Agilent Technologies Inc, Santa Clara, CA, USA). Targeted sequencing was carried out using Illumina sequencing and SureSelectXT Target Enrichment System (Agilent Technologies Inc., Santa Clara, CA, USA). 

### 4.4. Sequence Alignment and Variant Annotation

Burrows–Wheeler Aligner (BWA) was utilized to map the paired-end clean reads to the human reference genome (hg19). After the discovery of the genomic variant, the program ANNOVAR [53] was used to annotate the variants in a variety of ways, including the genomic regions impacted by the variants (RefSeq and Genecode), protein-coding changes and deleteriousness prediction (SIFT [54], PolyPhen [55] and MutationAssessor [56]), mRNA secondary structure (RNAfold) [24], allele frequency (1000 Human Genome) [57], disease associations (dbSNP [58], COSMIC (cancer.sanger.ac.uk) [59], OMIM [60], GWAS Catalog [61] and HGMD [62]) and pathway annotation (Gene Ontology [63], KEGG [64] and Reactome [65]).

### 4.5. Validation of Genomic Variants

The discovered somatic variants were then validated using the Sanger sequencing method. Primers were designed using the NCBI Primer Tool (National Center for Biotechnology Information, Bethesda MD, USA) and Primer3Plus [66]. The sequencing results were analyzed using the SnapGene Viewer 5.3.2. The primers used for validation were *IL12Rß2* rs2229546 5′-GCTGAGAGCAGACAACTGGT-3′ (forward), 5′-CCATCATGGGTGGGAAGGTC-3′ (reverse), rs2228420 5′-GGGCGCATACACCAATCAG-3′ (forward), 5′-TTTCCCTGACCCATGGCAG-3′, *IL23R* rs10889677 5′-TCTGTGCTCCTACCATCACC-3′ (forward), 5′-TGTGCCTGTATGTGTGACCA-3′ (reverse) and *JAK2* rs2230722 5′-GAGATCTTGCCATGTTGCCC-3′ (forward), and 5′-ACACTGCCATCCCAAGACAT-3′ (reverse).

### 4.6. Statistical Analysis

Normally distributed variables are presented as the mean ± standard deviation, and non-normally distributed variables as the median (25th and 75th percentiles). Statistical analyses were performed using SPSS 26 software (SPSS Inc., Chicago, IL, USA).

## 5. Conclusions

We were able to identify somatic variants in the PI3K-related genes among UC, CRC and CAC, and it was discovered that most of these variants were found in the *IL23R, IL12Rß1* and *IL12Rß2* genes, followed by *TYK2*, *JAK2* and *OSMR*. The discovery of *IL23R* variant rs10889677 as a possible mutation may help to give an insight on how the cytokine-induced PI3K pathway links inflammation with a higher risk of developing cancer, opening the door to improved care for CAC patients in the near future.

## Figures and Tables

**Figure 1 ijms-23-11472-f001:**
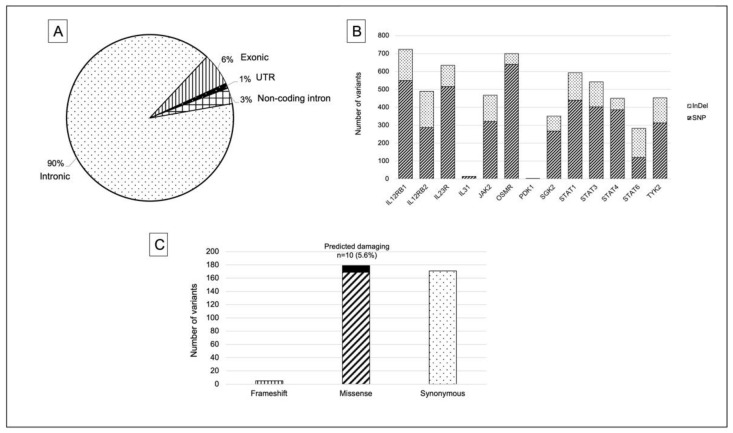
(**A**) Pie chart displaying the overall distribution of the variants identified in PI3K-related genes. (**B**) A bar column showing the total number of variants discovered in each PI3K-related gene and fractionated into Indel and SNPs. (**C**) Bar charts displaying the different somatic alterations found in the exonic region across all samples.

**Figure 2 ijms-23-11472-f002:**
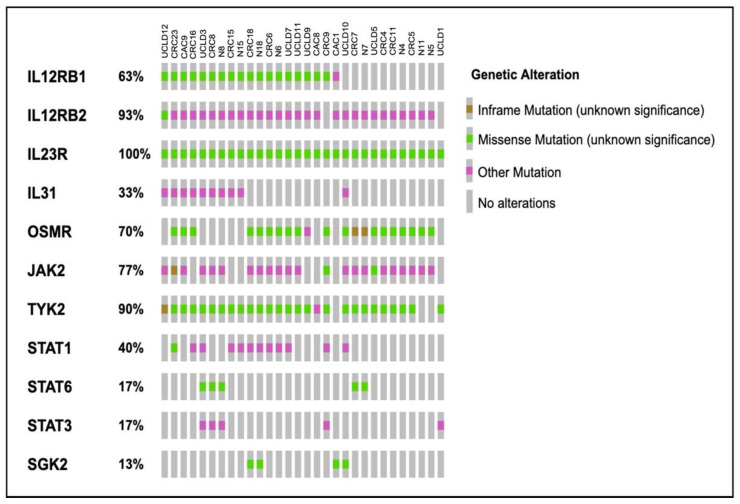
Oncoprint diagram showing the genetic alterations found in the exonic region of PI3K-related genes. Each bar represents the patient’s number [22,23].

**Figure 3 ijms-23-11472-f003:**
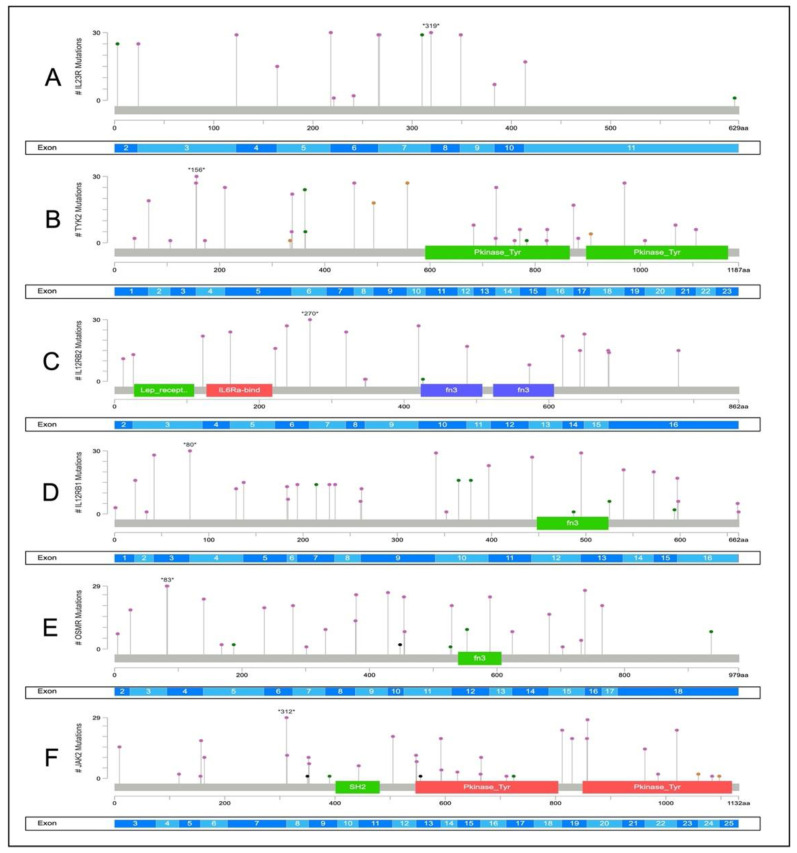
Distribution of somatic mutations within the functional domain of each gene. Circle and the hues green (missense), black (truncating mutation), and purple symbolize the mutations of other genes. The number with asterisk displays the location of protein change. The number of mutations identified in the coding area is shown by the length of the line. (**A**) *IL23R*, (**B**) *TYK2*, (**C**) *IL12Rß2,* (**D**) *IL12Rß1*, (**E**) *OSMR*, and (**F**) *JAK2* changes were identified.

**Figure 4 ijms-23-11472-f004:**
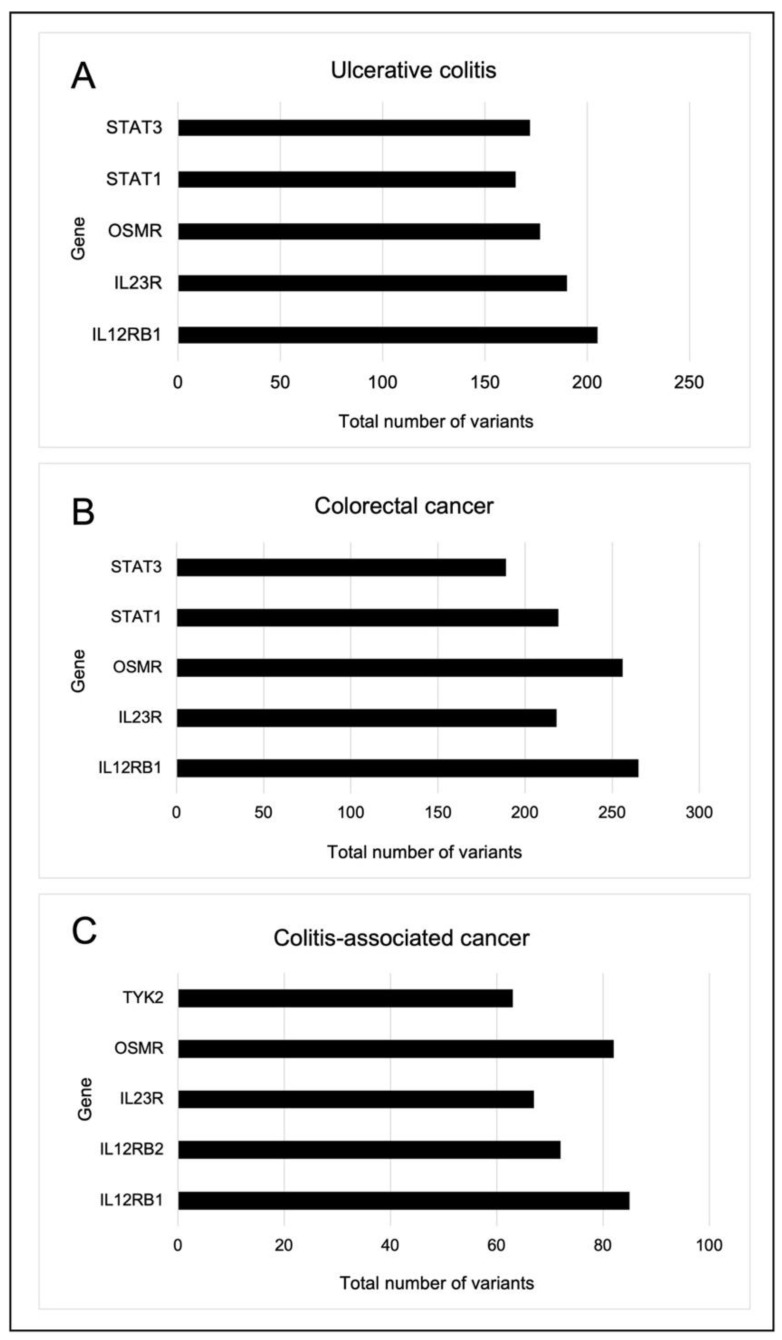
The top five altered genes for each group are shown in a bar graph. (**A**) Ulcerative colitis (**B**) Colorectal cancer, and (**C**) Colitis-associated cancer groups.

**Figure 5 ijms-23-11472-f005:**
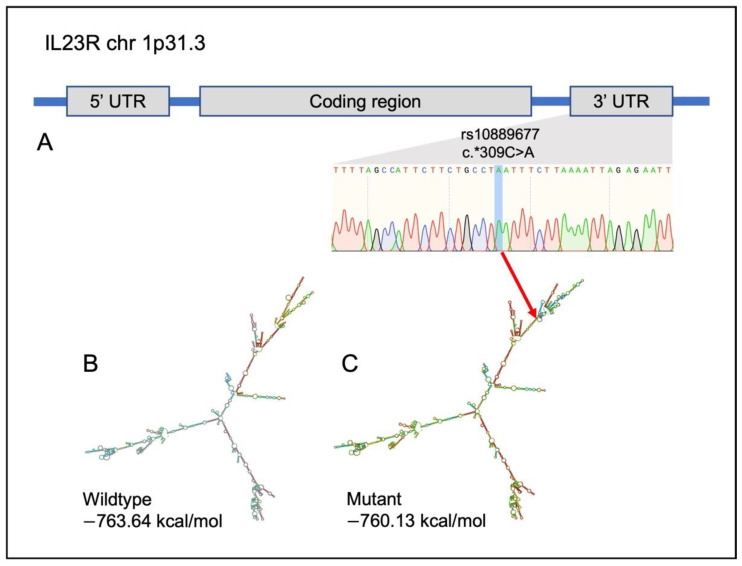
Details on *IL23R* variant rs10889677 and the in silico prediction of the mRNA secondary structure by RNAfold. (**A**) Location of the variant in the *IL23R*. (**B**) Wildtype (**C**) Mutant. Red arrow showing the location of nucleotide change [24].

**Table 1 ijms-23-11472-t001:** Clinical and demographic details of the recruited patients. All data are expressed as *n* except where indicated in the table. UC, ulcerative colitis; CRC, colorectal cancer; CAC, colitis-associated cancer; n, number.

	UC (n = 8)	CRC (n = 11)	Normal (n = 8)	CAC (n = 3)
Median age (range)	65.5 (60–69)	60 (36–74)	64 (51–74)	59 (20–69)
Race				
Malay	3	10	7	1
Chinese	3	1	1	-
Indian	2	-	-	2
Gender				
Male	4	5	3	-
Female	4	6	5	3
Smoking status				
Ex-smoker	-	1	1	-
Non-smoker	8	10	7	3
Stage	Not applicable		Not applicable	
I		4		-
II		3		-
III		4		3
Adenocarcinoma types	Not applicable		Not applicable	
Poorly differentiated		1		1
Moderately differentiated		9		-
Well differentiated		1		2
Mayo score (range)	1–3	Not applicable	Not applicable	Data unavailable
Geboes score (range)	2A.1–2A.2	Not applicable	Not applicable	Data unavailable

**Table 2 ijms-23-11472-t002:** List of recurrence somatic variants in two samples or more in each UC, CRC and CAC group. UC, ulcerative colitis; CRC, colorectal cancer; CAC, colitis-associated cancer.

Group	Gene	Location	dbSNP	Changes	Prediction
UC	*IL12Rß1*	Intronic	rs201422056	g.18174947_18174948del	Not applicable
	*IL12Rß2*	Intronic	rs17838042	g.67792801G>C	Not applicable
		Intronic	rs17129778	g.67787691A>T	Not applicable
		Intronic	rs17129794	g.67794918A>C	Not applicable
		Intronic	rs147756804	g.67796641_67796646del	Not applicable
	*IL23R*	Intronic	rs41313260	g.67706309C>T	Not applicable
	*SGK2*	Intronic	rs73620603	g.42195665C>T	Not applicable
	*OSMR*	Intronic	rs367864552	g.38881296_38881299del	Not applicable
		Intronic	rs55964556	g.38931022_38931024del	Not applicable
		Intronic	rs757333768	g.38881299ins	Not applicable
	*STAT4*	Intronic	rs370820216	g.191898876T>C	Not applicable
	*STAT6*	Intronic	-	g.57494483_57494499del	Not applicable
		Intronic	-	g.57494421ins	Not applicable
CRC	*IL12Rß1*	Exonic	rs370238890	c.1781G>A; p.G594E	Benign
	*IL12Rß2*	Intronic	-	g.67860953_67860954del	Not applicable
	*IL23R*	Intronic	rs767258696	g.67699612_67699616del	Not applicable
	*OSMR*	Intronic	rs113727379	g.38885647C>T	Not applicable
		Exonic	rs34675408	c.561T>G; p.H187Q	Benign
	*JAK2*	Intronic	rs3780378	g.5112288C>T	Not applicable
	*STAT4*	Intronic	rs35593987	g.191916526_191916527del	Not applicable
		Intronic	rs11272763	g.191992821ins	Not applicable
	*STAT6*	UTR5	rs71802646	g.57505072_57505076del	Not applicable
CAC	*IL12Rß2*	Intronic	Not available	g.67795960ins	Not applicable
	*TYK2*	Intronic	Not available	g.10477409_10477412del	Not applicable

**Table 3 ijms-23-11472-t003:** List of somatic variants from UC and CRC that co-exist with CAC group. UC, ulcerative colitis; CRC, colorectal cancer; CAC, colitis-associated cancer.

Group	Gene	Location	dbSNP	Change
CAC with UC	*IL12Rß1*	Intronic	rs372889	g.18173603T>C
		Intronic	rs439409	g.18193613A>G
		Intronic	rs382634	g.18187562G>A
		Intronic	rs17878594	g.18173513C>T
		Intronic	Not available	g.18179560_18179562del
	*IL12Rß2*	Intronic	rs12410480	g.67803994G>T
		Intronic	rs145598332	g.67833145ins
		Intronic	rs66726768	g.67795956_67795960del
	*IL23R*	Intronic	Not available	g.67672567_67672569del
	*OSMR*	Intronic	rs79215370	g.38882285C>T
		Intronic	rs137968159	g.38919267_38919270del
	*JAK2*	Intronic	rs10283730	g.5073289G>A
		Intronic	rs7865719	g.5082333A>G
		Intronic	rs138377711	g.5111358_5111359del
	*TYK2*	Intronic	rs12720294	g.10469699A>G
		Intronic	rs12720293	g.10470293A>G
		Intronic	rs143429818	g.10469743ins
	*STAT1*	Intronic	rs2066803	g.191839459C>A
		Intronic	rs41371944	g.191844745T>C
		Intronic	rs376961322	g.191844269_191844270
	*STAT3*	Intronic	rs9909659	g.40473835G>A
		Intronic	rs8081037	g.40499158C>T
	*STAT6*	Intronic	Not available	g.57494483ins
		Intronic	Not available	g.57494421ins
		Intronic	rs398019756	g.57494880_57494881del
CAC with CRC	*JAK2*	Intronic	rs9987451	g.5113452C>T
	*STAT4*	Intronic	Not available	g.191940749_191940750del

## Data Availability

The raw data in this study are available upon request.

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
