# Peer review of "Targeted Sequencing of Cytokine-Induced PI3K-Related Genes in Ulcerative Colitis, Colorectal Cancer and Colitis-Associated Cancer"

_ijms, 2022, doi:10.3390/ijms231911472_

Round 1
Reviewer 1 Report
This manuscript describes the identification of somatic variants in the cytokine-induced PI3K related genes in UC, colorectal cancer (CRC) and CAC from 30 biopsies using Illumina sequencing and SureSelectXT Target Enrichment System for targeted sequencing on PI3K related genes. Additional secondary structure prediction using RNAfold revealed that mutant structures were more unstable than wildtype structures.
A few concerns for the authors.
11. On page 3, in Table 1, it would be better to include the total number of patients in each group in the table even though the information is available somewhere else.
22. On page 3, Line 113-114, the authors indicated that “Long standing ulcerative colitis (n=8), colitis-associated cancer (n=3), colorectal cancer (n=11) and paired normal colorectal mucosa (n=8) made up the total 30 samples”, which means the total 30 samples include the normal colorectal mucosa samples. Then on page 4, from line 138 to line 140, “IL23R missense mutations more frequently than any other genes (30/30 samples). Then, IL12Rß2 (28/30 samples), TYK2 (27/30 samples), JAK2 (23/30 samples), OSMR (21/30 samples) and IL12Rß1 (19/30 samples) follow.” This information is confusing. Do the normal colorectal mucosa samples also have the same mutations as the UC, CRC and CAC samples?
33. On page 5, line 150-151, “Nearly half of the 314 total mutations, which appeared in two or more samples.” is a phrase, not a sentence. Need rephrase.
44. On page 6, line 171-172, “Another gene with a lot of modification numbers is OSMR, with 699 alterations, 281 which were recurrent mutations.” Should add a “of” before which.
55. The in silico prediction of the secondary structure IL23R variant (rs10889677) revealed that the mutant structure is less stable than the structure of wildtype. And on page 11, line 317-318, “Stability is essential in mRNA secondary structure because stable complexes may increase translation efficiency.” If the mutant is having decreased translation efficiency, how can this mutation affect the disease status?
66. Throughout the manuscript, when the authors were trying to present the numbers, no consistent format is used as sometimes they used Arabic numerals, sometimes they used English words. For examples, 1) page 5, line 151-152, “From the 634 total mutations, 57 were recurrent missense mutations, three were synonymous mutations, 551 were intronic and 23 were UTR.” 2) line 162-164, “IL12Rß1 had the highest number of alterations, with 723 mutations, including 299 recurrence while being affected in just 60% of all cases. Fifty-five recurrence missense mutations were found between exon 7 to 15…” and some more.
Author Response
Hi
We have revised and answer all the comments raised by the reviewer.
Thank you

Reviewer 2 Report
The authors have sequenced genes related to the cytokines-induced PI3K signaling pathway with the objective of identifying mutations in colitis-associated cancer, long-standing ulcerative colitis, and sporadic colorectal cancer patients.
Overall, the manuscript is clearly written. The experiments have been carefully designed and the results are nicely presented in appropriate figures and Tables. The main issue of this paper is that it is a very descriptive study with very little in terms of the mechanism and clinical relevance of the findings.
It is not clear the consequences of the described somatic alterations in terms of tumor initiation/promotion.
The prediction analysis for the IL23R variant (rs10889677) could be better explained in terms of the biological significance of these findings.
It was not clear to which mutations the authors used protein function prediction methods like SIFT and PolyPhen-2 and which were the results obtained using these methods.
In the Discussion section:
The authors can suggest ways future research might confirm the findings or take the research forward.
The discussion needs to address the limitations of the study.
Minor points
1) The quality of the English language of text should be revised.
2) In the Material and Methods section, the authors should add references for the programs used under subtitle 4.4. Sequence alignment and variants annotation, such as those pointed out by the authors: “SIFT, PolyPhen and MutationAssessor), mRNA secondary structure (RNAfold), allele frequency (1000 Human Genome), disease associations (dbSNP, COSMIC, OMIM, GWAS Catalog and HGMD) and pathway annotation (Gene Ontology, KEGG and Reactome).”
Author Response
Hi
We have addressed all the comments raised by you.
Thank you

Reviewer 3 Report
The article presented by Nurul Nadirah Razali titled “Targeted sequencing of cytokine-induced PI3K related genes in ulcerative colitis, colorectal cancer, and colitis-associated cancer” is well written, clear, and easy to read. The topic is very interesting, but unfortunately, the biopsies' numerosity is too small by groups to conclude on the new mutation found in the IL-23 gene. The authors should obtain more samples.
Author Response

(The authors gave the same response as above.)

Round 2
Reviewer 2 Report
Many thanks for your submission and response to the previous comments. There is improvement in the presentation of the manuscript, I have no further suggestions.